# Comp2Comp: Open-Source Body Composition Assessment on Computed Tomography

**Louis Blankemeier**[*1]                                    LOUIS.BLANKEMEIER@STANFORD.EDU

[1] *Stanford University, CA, USA*

**Malte Jensen**[*1]                                                   MEKJ@STANFORD.EDU

**Eduardo Pontes Reis**[*1,2]                                        EDREIS@STANFORD.EDU

[2] *Hospital Israelita Albert Einstein, Sao Paulo, Brazil*

**Juan Manuel Zambrano Chaves**[*1]                                       JMZ@STANFORD.EDU

**Adrit Rao**[1]                                                   ADRITRAO@STANFORD.EDU

**Sally Yao**[1]                                                  YAOHANQI@STANFORD.EDU

**Pauline Margaret Berens**[1]                                       PBERENS@STANFORD.EDU

**Andrew Wentland**[3]                                          ALWENTLAND@WISC.EDU

[3] *University of Wisconsin-Madison, WI, USA*

**Bhanushree Bahl**[4]                                    BHANUSHREE.BAHL@CARPL.AI

[4] *CARPL.ai, New Delhi, India*

**Kushboo Arora**[4]                                      KHUSHBOO.ARORA@CARPL.AI

**Oliver Oppers Aalami**[1]                                      AALAMI@STANFORD.EDU

**Bhavik Patel**[5]                                            PATEL.BHAVIK@MAYO.EDU

[5] *Mayo Clinic, MN, USA*

**Leon Lenchik**[6]                                       LLENCHIK@WAKEHEALTH.EDU

[6] *Wake Forest University, NC, USA*

**Marc H. Willis** [1]                                         MARC.WILLIS@STANFORD.EDU

**Robert D. Boutin**[1]                                            BOUTIN@STANFORD.EDU

**Arjun D. Desai**[*1]                                            ARJUNDD@STANFORD.EDU

**Akshay S. Chaudhari**[*1]                                      AKSHAYSC@STANFORD.EDU

**Editors:** Under Review for MIDL 2023

## Abstract

Computed tomography (CT) can provide quantitative body composition metrics of tissue volume, morphology, and quality which are valuable for disease prediction and prognostication. However, manually extracting these measures is a cumbersome and time-consuming task. Proprietary software to automate this process exist, but these software are closed-source, impeding large-scale access to and usage of these tools. To address this, we have built `Comp2Comp`, an open-source Python package for rapid and automated body composition analysis of CT scans. The primary advantages of `Comp2Comp` are its open-source nature, the inclusion of multiple tissue analysis capabilities within a single package, and its extensible design. We discuss the architecture of `Comp2Comp` and report initial validation results. `Comp2Comp` can be found at `https://github.com/StanfordMIMI/Comp2Comp`.

**Keywords:** computed tomography, segmentation, body composition, abdominal CT.

---

* Contributed equally or co-senior authorship

## 1. Introduction

Quantitative metrics from computed tomography (CT) can provide diagnostic and prognostic biomarkers for acute and chronic health conditions (Lee et al., 2022a; Thibault et al., 2012; Kuriyan, 2018). Such measures can provide a more objective evaluation of body composition (BC) than traditional clinical measurements (e.g. weight, body mass index (BMI), waist circumference, skinfolds) (Zeng et al., 2021). However, manually extracting quantitative BC measures from CT scans is time-consuming and prone to inter-reader variability, which considerably limits their utility in clinics and research studies.

We introduce `Comp2Comp`, an open-source Python package to expedite CT-based BC analysis. `Comp2Comp` contains methods to automatically segment CT images, extract quantitative BC measures, and generate polychromatic visual reports. `Comp2Comp` is designed to be extensible, enabling the development of complex clinically-relevant applications. The package is hosted on the GitHub platform at `https://github.com/StanfordMIMI/Comp2Comp` with a permissive license.

## 2. Inference Pipelines

A key component of `Comp2Comp` is its inference pipeline system. Inference pipelines string together sequences of building-block inference class modules which perform specific tasks like machine learning, saving or loading data, visualizing outputs, or other computational tasks. Furthermore, inference pipelines can be reused within other inference pipelines. This modular structure shortens iteration cycles for developing complex clinical applications. We list the pipelines currently implemented in `Comp2Comp`. Each of these pipelines saves numerical or categorical results, image reports, and segmentation files. All trained models are available on HuggingFace and downloaded automatically within `Comp2Comp`.

### 2.1. Spine Bone Mineral Density from 3D Trabecular Bone Regions at T12-L5

Retrospective studies have established that L1 trabecular bone density of $< 90$ Hounsfield units (HU) is associated with a high risk of vertebral fracture (odds ratio, 32) (Graffy et al., 2017; Lee et al., 2018). Large scale screening for osteoporosis using CT has been validated in retrospective cohorts (Roux et al., 2022; Pickhardt et al., 2020), but is not yet widely implemented because automated techniques have not been freely disseminated.

We provide options for running the TotalSegmentator (TS) spine model (Wasserthal et al., 2022) as well as an nnUNet trained on VerSe (Sekuboyina et al., 2021; Löffler et al., 2020; Liebl et al., 2021) and TS data (Wasserthal et al., 2022). We develop heuristics for extracting 3D regions of interest (ROIs) from vertebral body trabecular bone as described previously (Blankemeier et al., 2023). `Comp2Comp` reports average HUs within these ROIs.

We validate our method with TS on 40 contrast-enhanced CT scans from the Stanford emergency department. Comparing to central regions extracted from labeled T12-L5 vertebral bodies, we achieve an average HU percent error of 1.82±0.86 across these 6 levels.

### 2.2. Slice-by-Slice 2D Analysis of Muscle and Adipose Tissue

Sarcopenia, defined by the loss of muscle tissue and muscle function, is associated with adverse outcomes, such as post-operative complications (Papadopoulou et al., 2020; Surov

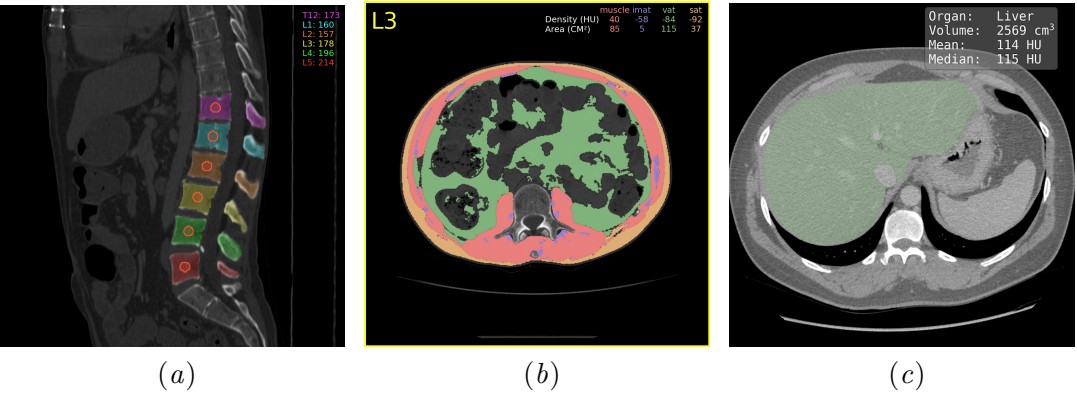

Figure 1: (a) Curved planar projection from our spine inference pipeline. Within the image, we report average ROI HU at T12-L5. (b) Output image from our spine, muscle, and adipose tissue pipeline. Here, the level is automatically determined using the spine model. Within the image, we report mean HU and area of each segmented tissue. (c) Output image from our liver, spleen, and pancreas inference pipeline. Within the image, we report the organ volume, as well as mean and median HU.

and Wienke, 2022). Adipose tissue, particularly visceral adipose tissue (VAT), is a modifiable risk factor for numerous medical conditions (Vilalta et al., 2022; Rao et al., 2021; Katzmarzyk et al., 2022).

We provide two models for 2D segmentation of muscle and adipose tissue. Both are 2D UNet models trained on axial CT slices at the L3 vertebral level.

On an internal test set of 40 abdominal contrast-enhanced cases at the L3 vertebral level, we achieve the following mean (standard deviation) Dice scores: 0.97 (0.03), 0.96 (0.05), and 0.97 (0.02) for muscle, VAT, and subcutaneous adipose tissue (SAT) respectively. The error in HU and area averaged below 1% and 2%, respectively, for all tissues. For the same three tissues, we achieve 94.7 (5.9), 94.6 (6.7), and 93.2 (14.8) on 20 external cases.

### 2.3. Contrast Phase Detection

Contrast agents are used to enhance the radiodensity of the blood vessels and vascularized tissues. Determining contrast phase is an important step for the successful application of algorithms with outputs that are sensitive to pixel intensity.

The Comp2Comp contrast phase detection pipeline consists of segmenting key anatomical structures, extracting metrics from these structures, and classifying these into one of 4 classes (non-contrast, arterial, venous and delayed) using a gradient boosting classifier.

On 362 internal test cases, our method achieves F1 scores of 0.96, 0.78, 0.92, and 0.95 for non-contrast, arterial, venous, and delayed phases respectively.

### 2.4. 3D Analysis of Liver, Spleen, and Pancreas

Liver disease and cirrhosis, the ninth leading cause of death, can be predicted by the volume, morphology, and attenuation of liver and spleen structures (Lee et al., 2022b). Volume measures can identify enlarged organs and aid in transplant planning (Linguraru et al., 2010).

Comp2Comp provides 3D analyses of the liver, spleen and pancreas. The volume, as well as the mean and median HU intensities are recorded. The organ segmentations are displayed in the axial and coronal planes. The slice with the largest cross-sectional area is displayed in the axial plane and the slice with the longest continuous length is displayed in the coronal plane. To segment these organs, we leverage TS (Wasserthal et al., 2022).

### 2.5. Combining Inference Pipelines: End-to-End Spine, Muscle, and Adipose Tissue Analysis at T12-L5

The modular design of Comp2Comp makes it easy to combine various Comp2Comp inference pipelines. Our spine, muscle, and adipose tissue pipeline combines the spine pipeline with the muscle and adipose tissue pipeline to analyze spine bone mineral density, muscle and adipose tissue at T12-L5.

To select the axial slices for muscle and adipose tissue segmentation, we compute per-level superior/inferior (SI) centers. Comparing to SI centers from our labeled T12-L5 vertebral bodies, we achieve an average error of 4.2±2.0mm across T12-L5. On 20 external cases, our segmentation model achieves mean (standard deviation) Dice scores, averaged across T12-L5, of: 0.88 (0.08), 0.93 (0.07), 0.91 (0.16) for muscle, VAT, and SAT respectively.

### 3. Conclusion

We present C2C, a tool for automated analysis of multiple tissues that is extensible and open source. We hope that Comp2Comp will increase the usage of BC analysis in large-scale research studies and clinical settings. We welcome any contributions from the community.

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
