# OpenReview forum: "Comp2Comp: Open-Source Body Composition Assessment on Computed Tomography"
_MIDL.io/2023/Short_Paper_Track — MIDL 2023 Short paper track Poster_

### Official Review · Reviewer_ykid · 2023-04-12
**Valuable open science effort**

**Rating:** 7
**Confidence:** 4

**Review:**

This paper presents an open-source package that enables the deep learning-based segmentation of various structures from CT images to perform body composition assessment. The authors briefly but clearly describe the models used for each application with the performance they reach.
Developing tools openly available to the community represents an important effort that is not always well rewarded. Presenting such work may encourage members of the MIDL community to join this effort.

---

### Official Review · Reviewer_ivpV · 2023-04-22

**Rating:** 9
**Confidence:** 5

**Review:**

The paper proposes an open source python library for body composition assessment of CT images. The paper is well organized and well written. The effort to open-source adipose tissue segmentation models is also appreciated.